# An Ocean between the Waves: Trends in Antimicrobial Consumption in Hospitalized Patients with COVID-19

**DOI:** 10.3390/antibiotics13010055

**Published:** 2024-01-05

**Authors:** Xavier Durà-Miralles, Gabriela Abelenda-Alonso, Alba Bergas, Júlia Laporte-Amargós, Enric Sastre-Escolà, Ariadna Padullés, Jordi Carratalà, Carlota Gudiol

**Affiliations:** 1Department of Infectious Diseases, Bellvitge University Hospital, Hospitalet de Llobregat, 08907 Barcelona, Spain; xavierduramiralles@gmail.com (X.D.-M.); abergas@bellvitgehospital.cat (A.B.); esastre@bellvitgehospital.cat (E.S.-E.);; 2Bellvitge Biomedical Research Institute (IDIBELL), Hospitalet de Llobregat, 08907 Barcelona, Spain; 3Center for Biomedical Research in Infectious Diseases Network (CIBERINFEC), Instituto de Salud Carlos III, 28029 Madrid, Spain; apadulles@bellvitgehospital.cat; 4Department of Pharmacy, Bellvitge University Hospital, Hospitalet de Llobregat, 08907 Barcelona, Spain; 5Facultat de Medicina i Ciències de la Salut, University of Barcelona, Campus Bellvitge, Hospitalet de Llobregat, 08907 Barcelona, Spain; 6Institut Català d’Oncologia (ICO), Hospital Duran i Reynals, IDIBELL, Hospitalet de Llobregat, 08908 Barcelona, Spain

**Keywords:** COVID-19, antibiotic, pandemic, stewardship

## Abstract

We assessed the antibiotic use in SARS-CoV-2-infected patients during four different waves of the COVID-19 pandemic, as well as its trends over the period and associated risk factors. We performed a cross-sectional retrospective analysis nested in a prospectively collected cohort of hospitalized adult patients with COVID-19 at a university hospital in Spain. A total of 2415 patients were included in this study, among whom 1120 corresponded to the first wave. The highest percentage of patients receiving some sort of antibiotic treatment was higher during the first wave (77.6%) than during the others; nevertheless, our calculation of the average DOT (days of antibiotic treatment) per 100 patient days of stay found that the highest antibiotic prescription rate corresponded to the second pandemic wave (61.61 DOT/100 patient days), which was associated with a higher ICU admission rate and a lower SpO_2_/FiO_2_ ratio at admission. After the second wave, the prescription rates presented a steady downward trend. With regard to the use of specific antibiotic families, amoxicillin/clavulanate was the most used antibiotic in our cohort (14.20 DOT/100 patient days) due to a high prescription rate during the first wave. According to the “AWaRe” WHO classification, antibiotics corresponding to the “Watch” group were the most prescribed (27.92 DOT/100 patient days). The antibiotic use rate fell progressively, but it remained high during all four waves analyzed. In conclusion, antibiotic use was high throughout all the waves that were analyzed, despite a relatively low incidence of bacterial coinfection and superinfection. Efforts should be made to keep antimicrobial stewardship programs active, especially in complicated epidemiological situations, such as the SARS-CoV-2 pandemic.

## 1. Introduction

For two years, from the end of 2019 up to 2021, the COVID-19 pandemic dominated every aspect of healthcare and diverted attention away from the advances and routine work in other important areas, such as antimicrobial resistance. During the initial days of this pandemic, the uncertainty and unfamiliarity with this new disease led clinicians to base their therapeutic decisions on previous experiences, such as the 2009 influenza pandemic, in which bacterial coinfection was frequent and an important predictor of poor outcomes [1]. Those earlier findings initially led the World Health Organization (WHO) to recommend the use of empirical antibiotics in hospitalized patients with COVID-19 pneumonia [2].

Although this initial recommendation was later withdrawn when it emerged that the prevalence of bacterial co-infection in these patients was low, certain factors may have contributed to an unwarranted increase in antimicrobial use during the COVID-19 pandemic [3]. On the one hand, due to the huge additional workload caused by the pandemic, many antimicrobial stewardship tasks were neglected, albeit unintentionally. On the other hand, the sub-optimal hand hygiene practices at a time of the mass use of personal protective equipment (PPE) and double gloves, along with the incorporation of untrained personnel in medical wards and intensive care units, may have favored the spread of hospital-acquired infections [4,5].

The Center for Disease Control and Prevention reported that US healthcare professionals prescribed 211.1 million antibiotic prescriptions in 2021, equivalent to 636 antibiotic prescriptions per 1000 individuals [6]. During the same year, the total average consumption of antibiotics in Europe was 15.0 defined daily doses (DDDs) per 1000 inhabitants per day, and in Spain, it was 18.49 DDDs per 1000 inhabitants per day [7].

Although some studies have reported an increased use of antimicrobials during the COVID-19 pandemic [8,9], few data are available on the trends of antimicrobial consumption during the different waves of this pandemic. In this study, we aimed to assess the overall prescription of antimicrobials during the COVID-19 pandemic at a university referral hospital and to compare their use during the different pandemic waves.

## 2. Results

A total of 2,415 adult patients were admitted to the hospital during the four waves. The first wave presented the highest number of hospitalizations (1120 patients, 46%), followed by the fifth wave (578 patients, 24%).

The distribution of patients admitted to the hospital during the different waves is shown in Figure 1. The first pandemic wave showed the sharpest increase and decrease in new hospital admissions, with subsequent waves presenting a flatter outline.

Table 1 displays the most relevant characteristics of the different waves. The first wave was the one with the shortest median hospital stay (8 days), whereas it accounted for the highest median of new hospitalizations per day (23 hospitalizations per day). The maximum peak of new hospitalizations through the entire pandemic was observed on 27 March 2020 with 92 new admissions. Intensive care unit (ICU) admission was significantly lower during the first wave (11.6%) compared to the second wave (15.8%) (OR 1.46 [1.03–2.07]). The highest ICU admission rate occurred during the third wave (16.5%), although without significant differences compared to the second wave (15.8% vs. 16.5% OR 1.05 [0.7–1.56]). The median ICU stay was shorter during the 5th wave (8 days), although these differences did not reach statistical significance.

The comparison of the different waves showed significant differences in patients’ baseline characteristics (Table 2).

During the fifth wave, the patients were younger, had fewer comorbidities, and were more likely to require intensive care unit (ICU) admission (79.3%). Fever at admission was more frequent during the first and second waves (22.7% and 23.4%, respectively) than in the third and fifth waves (14% and 14.6%, respectively). Overall, patients that were admitted during the first wave presented less severe pneumonia according to the SpO_2_/FiO_2_ ratio, which was >350 in 80.2% of the patients and <150 in 6.8%. Overall, the rates of microbiological testing were significantly higher during the first wave, and the microbiological yield was highest during the fifth. The mortality rate was lowest during the fifth wave (11.9%) and highest during the second (21.1%).

Regarding overall antibiotic consumption, 59% of patients received at least one antibiotic regimen during their hospitalization (Figure 2).

Figure 3 shows the percentages of patients receiving some sort of antibiotic prescription during the waves, which fell sharply after the first wave (from 77.6% to 42.9% during the second wave (OR 4.61 [3.55–5])). The great majority of patients received only one antibiotic regimen, and this remained the case throughout the study period.

Analyzing the primary outcome, the average antibiotic use rose to 53.29 DOT/100 patient days. Differences were observed between the waves, with a downward trend observed in antibiotic use between the first and second halves of the pandemic (Figure 4). Notably, a significant increase was observed between the first and second waves (58.71 DOT/100 patient days vs. 61.61 DOT/100 patient days, respectively; *p* < 0.01), with the second wave being the one with the highest antibiotic use.

Figure 5 displays the use of different antibiotic groups. Overall, the most used antibiotic during the four waves of this pandemic was amoxicillin/clavulanic acid with 14.20 DOT/100 patient days, followed by piperacillin/tazobactam with 8.61 DOT/100 patient days. Carbapenem use was 5.02 DOT/100 patient days.

The comparison between waves (Figure 6) showed a sharp drop in the use of amoxicillin/clavulanic acid after the first wave, when piperacillin/tazobactam remained the most frequently prescribed antibiotic (except for the second wave, where the use of cotrimoxazole reached a peak). Carbapenem use was slightly higher during the second and third waves compared to the first and the fifth. Macrolides were frequently used during the first wave, but their use fell sharply during the following periods.

Regarding the WHO’s AWaRe classification of antibiotics, the “Watch” category was the most used through all the waves, with a rate of 27.92 DOT/100 patient days, followed by the “Access” and “Reserve” categories, with an average use of 19.43 and 5.29 DOT/100 patient days, respectively (Figure 7).

This distribution was maintained over the course of the different waves (Figure 8).

## 3. Discussion

In the present study, we found a high overall rate of antibiotic intake, with more than half of the patients receiving some sort of antibiotic treatment during their hospital stay, and an average of 53.29 DOT/100 patient days. In a multicenter study conducted in Iran, Salehi et al. found an average use of 121.6 DDD/100 hospital bed days during the first six months of the pandemic [10]. Interestingly, 74% of patients in the meta-analysis by Chedid et al. received some sort of antibiotic therapy during the first wave of the pandemic [11], a rate similar to the figure of 77.6% of patients under an antibiotic treatment during the first wave in our cohort.

Many factors are potentially involved in this high antibiotic intake. For instance, when analyzing the first wave, the lack of experience with a novel virus that was proven to have a higher mortality rate than seasonal influenza [12] may well have had a major influence on the decision to implement empirical antibiotic treatments (after the initial alert in China, Spain was after Italy the second country in Europe to impose a lockdown due to the high number of cases [13]. During the first weeks of the pandemic, a hospital protocol based partially on the initial WHO recommendations [2] was implemented. According to this protocol, patients presenting with SARS-CoV2 pneumonia were recommended to be treated with amoxicillin/clavulanate combined with azithromycin (which was believed to favor patient outcomes, though this was later proven to be wrong [14]). Levofloxacin was recommended for patients with a beta-lactam allergy.

The high average antibiotic use found in our study seems to not be proportioned with the relatively low coinfection and superinfection rates reported in several studies during the pandemic. For instance, the meta-analysis by Lansbury et al. reported a coinfection rate of 7% [15], and the cohort study by Garcia-Vidal et al., performed in a very similar setting to our study, found that 7.2% of patients had a coinfection and/or superinfection during their hospital admission [16].

In the second wave of the pandemic, the percentage of patients receiving some sort of antibiotic treatment fell significantly, in agreement with Fjellveit et al. [17], who also reported a reduction in early antibiotic prescription. These findings probably reflect clinicians’ awareness of the low coinfection rate at admission. Following the withdrawal of macrolides and amoxicillin/clavulanate from the hospital protocol, a decrease in the use of these drugs was observed. Nevertheless, when considering the primary endpoint (DOT/100 patient days), overall antibiotic use turned out to be higher than during the first wave, mostly due to a significant increase in the use of cotrimoxazole and a moderate increase in the use of beta-lactams and fluoroquinolones.

There are three likely reasons for the increase in the use of these antibiotic families. The first is the lower average SpO_2_/FiO_2_ ratio at admission; this was probably due to more restrictive hospital admission criteria, which has been described as a driver of antibiotic prescription [8]. The second is the increase in the ICU admission rate, with more ventilator-associated respiratory infections and other nosocomial infections. The third is the extended use of glucocorticoids as part of COVID-19 pneumonia treatment [18], particularly in critically ill patients [19], which may have increased the risk of infection and was also associated with the prophylactic use of cotrimoxazole. The second wave also turned out to be the one with the highest use of the “Watch” and “Reserve” types of antibiotics, probably for the reasons proposed above.

However, other factors may also have played a role in the fall in average antibiotic use observed during the fifth wave: for instance, younger patients’ age, fewer patients presenting with a fever, lower need for mechanical ventilation, lower mortality rate, and the complete vaccination status of patients >65 years [20]. Interestingly, during the fifth wave, an increase in the use of “Access” antibiotics was observed, while “Watch” and “Reserve” antibiotics continued to fall; this was probably due to the progressive reestablishment of the antibiotic stewardship programs, the growing concern with the overuse of antibiotics during the previous waves, and the expertise accumulated by treating physicians after more than a year’s experience of the pandemic.

Despite its strengths, our study has some limitations that should be acknowledged. First, some information may have been lost due to the retrospective analysis of the prospectively collected data, and we may not have adequately controlled for certain confounders. Second, this study was conducted at a single center, and the results across other geographical areas with different healthcare practices may differ. Lastly, we were unable to provide data regarding adverse events related to antibiotic use or data showing the evolution of antimicrobial resistance, which would have offered information of particular interest.

In conclusion, antibiotic use was high throughout all the waves of the pandemic analyzed, despite a relatively low incidence of bacterial coinfection and superinfection. A progressive decrease in antibiotic use was observed during the last waves analyzed, in line with the gradual acquisition of knowledge regarding the novel viral infection and the improvement in the survival rate. Nevertheless, thorough antibiotic stewardship programs are warranted in order to avoid unnecessary antibiotic intake and to minimize the risk of antibiotic resistance in a pandemic setting.

## 4. Materials and Methods

### 4.1. Setting and Study Design

We performed a cross-sectional retrospective analysis nested in a prospectively collected cohort of hospitalized adult patients with COVID-19 at a university teaching hospital in Spain. Bellvitge University Hospital is a 700-bed hospital that serves as a public referral center for a population of roughly a million people in Catalonia, Spain. By 31 August 2021, this hospital had attended over 2500 adult patients hospitalized for COVID-19, all of whom presented a microbiologically proven SARS-CoV-2 infection with a positive nasopharyngeal or oropharyngeal polymerase chain reaction (PCR) test. Patients who did not present a microbiologically proven SARS-CoV-2 infection, even when COVID-19 was suspected, were excluded. Other exclusion criteria were patients < 18 years old and those who did not require hospitalization.

### 4.2. Data Collection and Outcomes

Data were collected in four time sections corresponding to the first, second, third, and fifth waves of the COVID-19 pandemic in our area. Although the definition of a pandemic wave is not well established and may vary according to different scientific and institutional opinions, for the current study, the different pandemic waves were defined according to the criteria of the Spanish health authorities. In the definition provided by the authorities, the separation between waves was established through the inflexion point on the 14-day national cumulative incidence of COVID-19 cases, after which a new increase in cases was detected [21]. For the convenience of the research, the entire period included in each wave was not included in the analysis, but only the weeks with the highest occupancy of hospital beds were included. The first wave included patients hospitalized between 9 March and 15 April 2020; the second, from 1 October to 30 November 2020; the third, from 1 January to 28 February 2021; and the fifth, from 1 July to 31 August 2021. The wave considered the “fourth” by the Spanish health authorities, recorded during the spring of 2021, presented a very low incidence of cases; so, data from this period were not analyzed in this study. All patients were followed up until their in-hospital death or hospital discharge.

The primary outcome of our study was the antimicrobial use during the different COVID-19 waves. According to the World Health Organization Collaborating Center for Drug Statistics Methodology, the defined daily dose (DDD) is the gold standard tool for monitoring and comparing drug use [22]. A denominator is ideally added in some health contexts, allowing for comparisons across various time periods and population groups (DDD per 1000 bed days). We chose a variation of this indicator, days of therapy (DOT) per 100 patient days, due to the high number of patients that needed ICU admission in our cohort and the lack of information regarding antibiotic dosages [23]. We defined the duration of treatment as the number of consecutive days during which a patient received a specific antimicrobial, and DOT as the aggregate sum of all the days during which a patient received any antibiotic. Over the course of these waves, we compared the number of patients receiving any antibiotic treatment, the number of antibiotic treatments received, and antibiotic treatment duration expressed as DOT per 100 patient days. We also compared the use of different antibiotics according to their family and potential ecological impact. To this end, we used the 2021 update of the AWaRe classification of antibiotics developed by the WHO Expert Committee of Selection and Use of Essential Medicines as a tool to support antibiotic stewardship efforts at local, national, and global levels [24]. This classification comprises three groups: Access, Watch, and Reserve, taking into account the impact of different antibiotics and antibiotic classes on antimicrobial resistance to emphasize the importance of their appropriate use [25]. The “Access” category includes beta-lactam + beta-lactamase inhibitor without antipseudomonal activity, sulfonamides, penicillins, and aminoglycosides. The “Watch” category includes antipseudomonal beta-lactam + beta-lactamase inhibitors, second-, third-, and fourth-generation cephalosporines without beta-lactamase inhibitor association, carbapenems, fluoroquinolones, and macrolides. The “Reserve” category includes lipopeptides, polymyxins, third-generation cephalosporines + beta-lactamase inhibitor association, fifth-generation cephalosporines, and oxazolidinones.

### 4.3. Statistical Analysis

Continuous and categorical variables were presented as medians (interquartile range) and absolute numbers (percentage), respectively. The Kolmogorov–Smirnoff test was used to evaluate normality, and the Mann–Whitney U-test, Chi-squared test, and Fisher’s test were used to compare differences between qualitative variables.

Differences in antimicrobial consumption between the different COVID-19 waves expressed in DOT/100 patient days were analyzed using the exact ratio test, achieving their corresponding confidence intervals (Cis) and *p-*values for each comparison. For all statistical analyses, 95% CIs were calculated. *p*-values of <0.05 were considered statistically significant. A two-tailed distribution was assumed for all *p*-values. The analysis was performed using IBM SPSS Statistics 29.0.

## Figures and Tables

**Figure 1 antibiotics-13-00055-f001:**
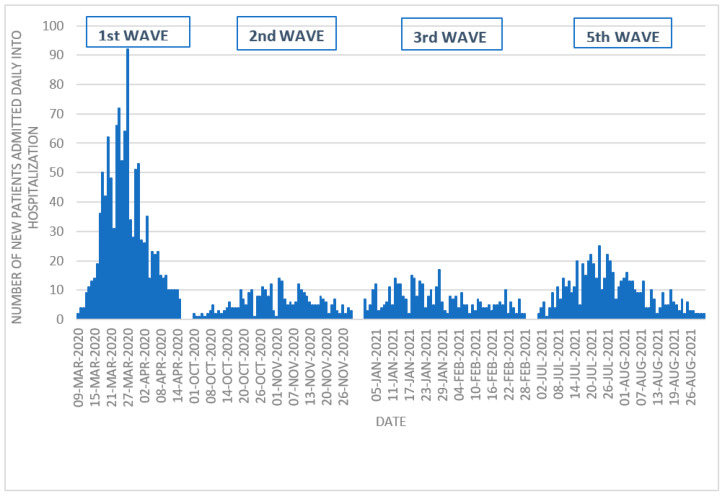
Distribution of patients admitted to the hospital during the different waves.

**Figure 2 antibiotics-13-00055-f002:**
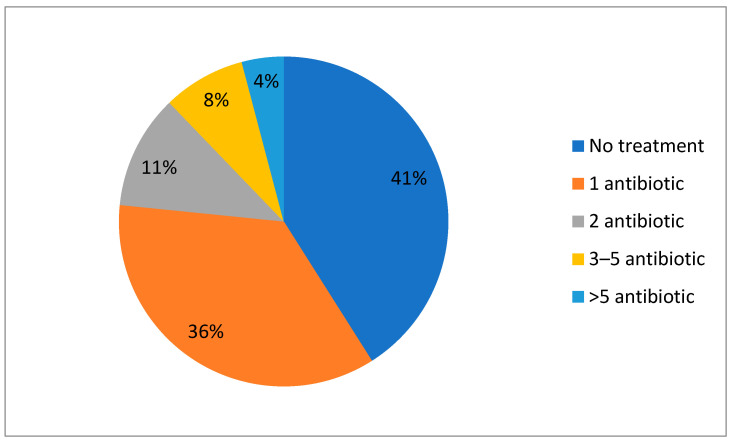
Overall antibiotic prescription during the four COVID-19 waves.

**Figure 3 antibiotics-13-00055-f003:**
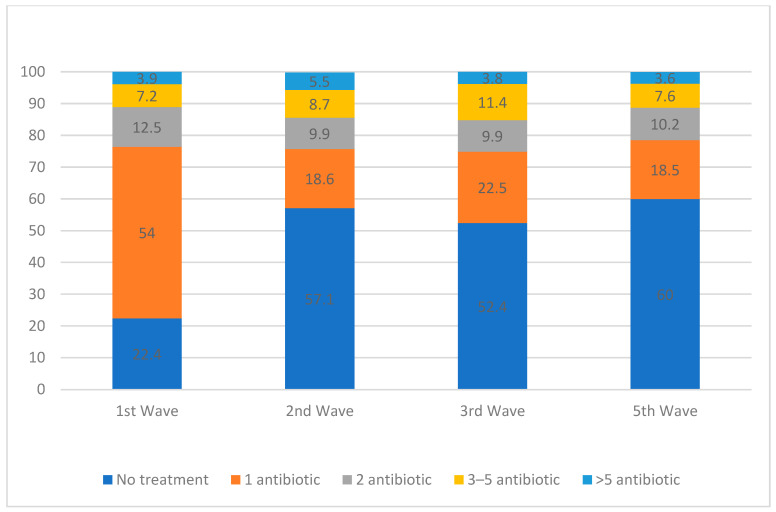
Percentage of patients receiving an antibiotic treatment during the COVID-19 waves.

**Figure 4 antibiotics-13-00055-f004:**
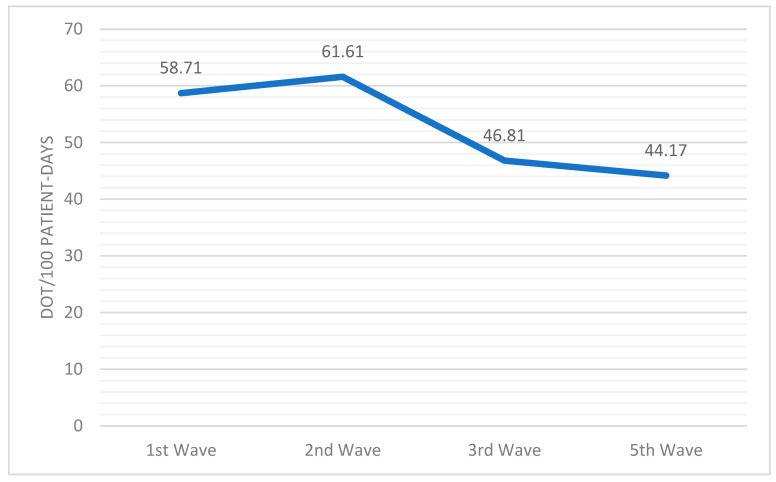
Evolution of overall antibiotic use along the waves.

**Figure 5 antibiotics-13-00055-f005:**
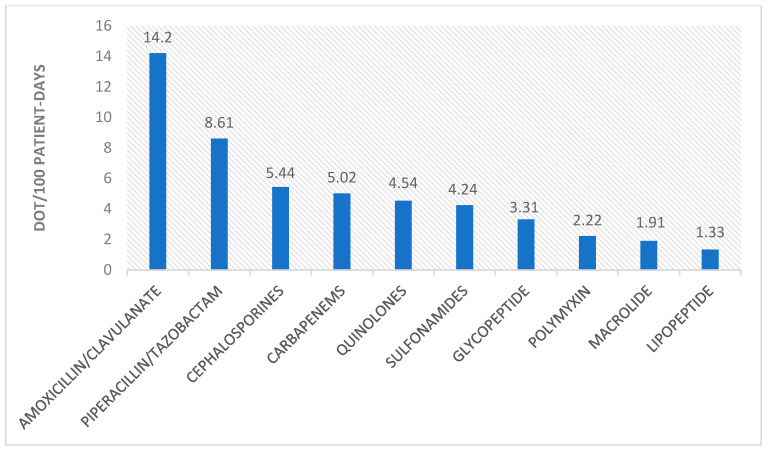
Overall average antibiotic family use.

**Figure 6 antibiotics-13-00055-f006:**
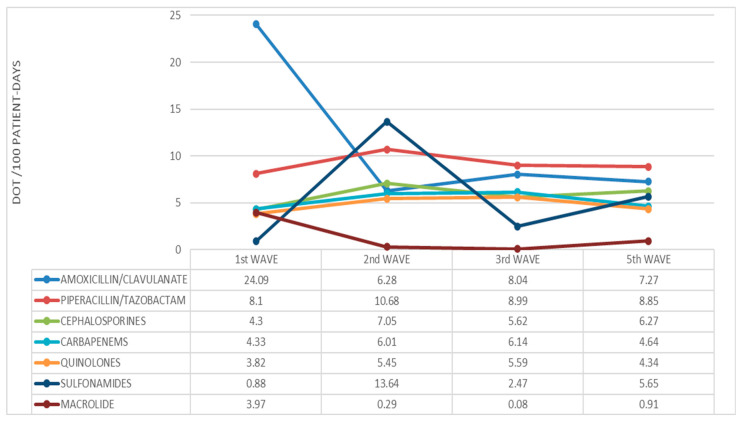
Evolution of average antibiotic family use during the waves.

**Figure 7 antibiotics-13-00055-f007:**
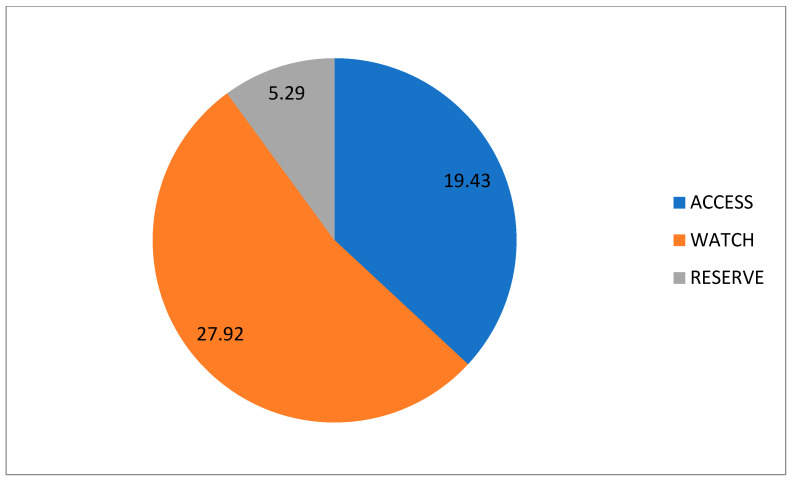
Overall average DOT/100 patient days’ antibiotic use according to the AWaRe classification.

**Figure 8 antibiotics-13-00055-f008:**
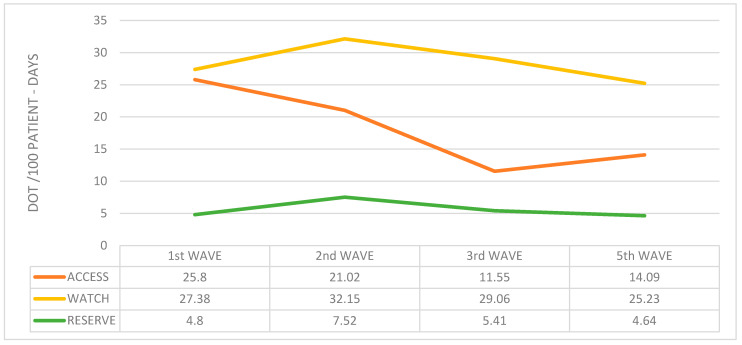
Average antibiotic use during the waves according to the AWaRe classification.

**Table 1 antibiotics-13-00055-t001:** Description of the different COVID-19 waves.

	1st Wave	2nd Wave	3rd Wave	5th Wave	*p*-Value
Dates	9 March 2020 15 April 2020	1 October 2020 30 November 2020	1 January 2021 28 February 2021	1 July 2021 31 August 2021	
Number of patients admitted	1120	322	395	578	
Length of hospital stay, median (range)	8 (1–138)	10 (1–198)	10 (1–247)	9 (1–123)	<0.001
Hospital admissions per day, median	23	5	6	9	<0.001
Maximum hospital admissions per day, n	92 (27 March 2020)	14 (2 November 2020)	17 (28 January 2021)	25 (23 July 2021)	
ICU admissions, n (%)	130 (11.6)	51 (15.8)	65 (16.5)	86 (14.9)	0.035
Days of ICU stay, median (range)	14 (1–93)	16 (1–107)	15 (1–153)	8 (1–109)	0.191

ICU: intensive care unit.

**Table 2 antibiotics-13-00055-t002:** Baseline characteristics, clinical features, and outcomes of patients hospitalized with a SARS-CoV-2 infection compared by waves.

	1st Wave n = 1120	2nd Wave n = 322	3rd Wave n = 395	5th Wave n = 578	*p*-Value
Male sex (%)	676 (60.4)	212 (65.8)	228 (57.7)	362 (62.6)	0.126
Age, years (median, range)	67 (22–98)	67 (26–94)	68 (26–101)	57 (20–99)	<0.001
Comorbidities: Diabetes mellitus, n (%) COPD, n (%) Charlson comorbidity index, avg	267 (23.8) 197 (17.6) 3.37	75 (23.3) 67 (20.8) 3.71	97 (24.6) 88 (22.3) 3.73	118 (20.4) 123 (21.3) 2.67	<0.001
Ceiling of care, n (%) Conventional oxygen device NIMV–HFNC IMV–ICU admission	221 (29.7) 158 (21.3) 364 (49)	38 (15.3) 52 (21) 158 (63.7)	29 (7.4) 79 (20.2) 284 (72.4)	41 (7.2) 77 (13.5) 452 (79.3)	<0.001
Fever at admission, n (%)	253 (22.7)	75 (23.4)	55 (14)	84 (14.6)	<0.001
SpO_2_/FiO_2_ at admission, n (%) >350 [300–349] [150–299] <150	891 (80.2) 61 (5.5) 83 (7.5) 76 (6.8)	202 (64.1) 27 (8.6) 42 (13.3) 44 (14)	231 (59.1) 29 (7.4) 73 (18.7) 58 (14.8)	332 (58.1) 56 (9.8) 121 (21.2) 62 (10.9)	<0.001
Shock at admission, n (%)	8 (0.7)	0	2 (0.5)	0	
Leucocytosis at admission, n (%)	167 (15)	53 (16.5)	77 (19.6)	113 (19.7)	0.049
ICU admission, n (%)	130 (11.6)	51 (15.8)	65 (16.5)	86 (14.9)	0.035
ADRS, n (%)	518 (46.3)	184 (57.1)	242 (61.3)	292 (50.5)	<0.001
Shock, n (%)	34 (3)	19 (5.9)	23 (5.8)	15 (2.6)	0.006
Exitus, n (%)	215 (19.2)	68 (21.1)	81 (20.5)	69 (11.9)	<0.001
*L. pneumophila* urinary antigen, n (%) Positive, n (%)	200 (17.9) 2 (1)	33 (10.3) 0	29 (7.4) 0	42 (7.3) 2 (4.8)	<0.001
*S. pneumoniae* urinary antigen, n (%) Positive, n (%)	217 (19.4) 15 (6.9)	39 (12.1) 6 (15.4)	33 (8.4) 4 (12.1)	44 (7.6) 6 (13.6)	<0.001
Blood culture, n (%) Positive, n (%)	433 (38.8) 12 (2.8)	133 (41.6) 4 (3)	123 (31.2) 5 (4.1)	126 (21.8) 4 (3.2)	<0.001
Ventilator-associated pneumonia, n (%)	29 (2.6)	21 (6.5)	13 (3.3)	26 (4.5)	0.006
Non-ventilator-associated pneumonia, n (%)	14 (1.3)	8 (2.5)	13 (3.3)	24 (4.2)	0.002
Nosocomial tracheobronchitis, n (%)	22 (2)	12 (3.7)	0	5 (0.9)	<0.001

COPD: chronic obstructive pulmonary disease; NIMV–HFNC: non-invasive mechanical ventilation–high-flow nasal cannula; IMV; intensive mechanical ventilation; ICU: intensive care unit; and ADRS: acute distress respiratory syndrome.

## Data Availability

The datasets generated and/or analyzed during the current study are not publicly available to preserve the individual privacy of the participants, but they are available from the corresponding author.

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
