# Peer review of "An Ocean between the Waves: Trends in Antimicrobial Consumption in Hospitalized Patients with COVID-19"

_antibiotics, 2024, doi:10.3390/antibiotics13010055_

Round 1
Reviewer 1 Report
Comments and Suggestions for Authors
1. Durà-Miralles et al. performed a single-center cross-sectional study that analyzed antibiotic use in SARS-CoV-2 hospitalized patients across four waves of the COVID-19 pandemic. In their study, they identified and compared the percentage of patients who received antibiotics, with the primary outcome being the average days of antibiotic treatment per 100 patients. This is an interesting study. However, it is limited by being a single-center study, and therefore the findings are not generalizable. The authors need to put the findings into a more global context. Specifically, how the study shall appear to a reader from another city/country.
2. The authors may need to indicate whether they have established any exclusion criteria for the sample.
3. Line 221: “The antimicrobial use during the different COVID-19 waves, expressed in days of therapy (DOT) per 100 patient-day”. Is this a commonly used endpoint (if so, please cite the proper references), or one that was established by the authors?
Comments on the Quality of English Language
I urge the authors to reread the manuscript carefully and correct any writing error. Examples below:
Please correct the typos: Line 16 “doe to a high prescription rate”. Line 154: “Levofloxacin was recommended of in patients”.
Reviewer 2 Report
Comments and Suggestions for Authors
Dear authors,
Congratulations on your work and hard work. The paper has high potential to garner interest from microbiologists, public health epidemiologists, and infectologists in the region and beyond. However, the paper needs to be improved, especially methodological descriptions needs to be enhanced.
1. Please add definition of wave or better add a pictorial histogram/time series showing the waves of patients in the hospitals.
2. Also can add a table regarding the wave characteristics like number of days of the wave, highest and lowest hospitalizations per day, cumulative hospitalizations, average number of days of hospitalizations, lowest and highest ICU admissions to help understand the differences in waves better.
Comments on the Quality of English LanguageEnglish is fine. Minor grammatical corrections are needed.
Round 2
Reviewer 1 Report
Comments and Suggestions for Authors
Accept.